## RESEARCH ARTICLE

# Characteristics of transferrin saturation and anemia-related biomarkers in patients with uterine adenomyosis

**Mari Kawamata, Fumitake Ito** ⓘ *, **Nanami Tahara, Akiyo Kakibuchi, Kazuya Yabumoto** ⓘ, **Yuko Izumi, Akihisa Katayama, Taisuke Mori**

Department of Obstetrics and Gynecology, Kyoto Prefectural University of Medicine, Graduate School of Medical Science, Kyoto, Japan

* fitoh@koto.kpu-m.ac.jp

## Abstract

### Background

Ferritin may be elevated as an acute-phase reactant in inflammatory conditions, potentially masking iron deficiency. We aimed to evaluate the frequency of iron deficiency and the diagnostic significance of transferrin saturation (TSAT) in women with uterine adenomyosis.

### Methods

This retrospective study included non-menopausal women aged 30–55 years who had iron metabolism markers measured at first visit. Anemia was defined as hemoglobin (Hb) <11 g/dL. Iron deficiency was defined using three criteria: ferritin-only (serum ferritin <20 ng/mL), TSAT-only (TSAT <20%), and a combined criterion (ferritin <20 ng/mL and/or TSAT <20%). Functional iron deficiency was defined as TSAT <20%, with ferritin ≥20 ng/mL. We compared the frequencies of anemia and iron deficiency across groups and evaluated correlations between Hb and TSAT (Pearson correlation using log-transformed TSAT; Spearman correlation as sensitivity analysis).

### Results

Iron deficiency was more common than anemia in all patient groups. In the adenomyosis group, the frequency of iron deficiency based on the combined criterion (65.5%) was significantly higher than that diagnosed using ferritin alone (31.0%). In the myoma and no uterine structural abnormality groups, iron deficiency frequency was similar regardless of the diagnostic criteria. The correlation between Hb and log(TSAT) was weak in the adenomyosis group (r = 0.2123, p = 0.1771) but strong in the myoma (r = 0.5465, p < 0.0001) and no uterine structural abnormality groups (r = 0.6945, p < 0.0001). Spearman analyses showed the same qualitative pattern (adenomyosis ρ = 0.358, p = 0.020; myoma ρ = 0.576, p < 0.0001; no uterine structural

**Data availability statement:** The de-identified dataset supporting the findings of this study has been deposited in Zenodo and is publicly available at: [10.5281/zenodo.18637560].

**Funding:** This work was partially supported by Grants-in-Aid for Scientific Research 21K09546 from the Ministry of Education, Culture, Sports, Science, and Technology (Japan). The funder had no role in study design, data collection and analysis, decision to publish, or preparation of the manuscript.

**Competing interests:** The authors have declared that no competing interests exist.

abnormality $\rho = 0.484$, $p = 0.0004$). Agreement between ferritin- and TSAT-based classifications was fair (Cohen's $\kappa = 0.375$; PABAK $= 0.371$), and a multivariable model for TSAT <20% with ferritin ≥20 ng/mL showed modest discrimination (AUC $= 0.678$).

## Conclusions

Measuring TSAT and ferritin levels enhances the detection of iron deficiency in women with adenomyosis and may help identify functional iron deficiency when ferritin interpretability is limited by inflammation.

---

## Introduction

Adenomyosis is a benign gynecological disorder characterized by the development of endometrial glandular epithelium and endometrial stromal tissue within the myometrium. The frequency of adenomyosis, as estimated from histopathological examinations after hysterectomy, varies widely, ranging from 8.8% to 61.5% [1,2]. The frequency is reported to be highest in patients in their early 40s [3]. Adenomyosis is associated with hypermenorrhea [4], dysmenorrhea [5,6], and obstetric complications such as preterm labor, preterm premature rupture of membranes, spontaneous abortion, gestational diabetes mellitus, small for gestational age, and preeclampsia [5]. The definitive treatment for adenomyosis is total hysterectomy; other treatments include endocrine therapies, such as low-dose estrogen-progestin, progestin, and gonadotropin-releasing hormone antagonist/agonist, and symptomatic treatments such as iron therapy and analgesics [7]. The management of anemia, a significant pathology of the disease, is particularly important.

Anemia is typically identified based on hemoglobin (Hb) levels; however, a subtype of iron deficiency can occur without overt anemia. Iron is essential for erythropoiesis and oxygen transport, and iron deficiency can cause symptoms such as fatigue, impaired concentration, dizziness, and headache, regardless of anemia [8,9]. Iron deficiency affects many essential biological processes such as DNA synthesis and repair and enzyme activity [10,11]. In premenopausal women, blood loss due to regular menstruation causes iron deficiency. However, iron deficiency often remains unrecognized, even in women routinely screened for anemia.

Iron deficiency presents as either absolute or functional deficiency. Absolute iron deficiency is characterized by reduced systemic iron levels, whereas functional iron deficiency is characterized by normal or higher systemic iron levels, uneven distribution of iron, and inadequate supply of iron to target tissues [2,12,13]. Absolute iron deficiency is diagnosed using low serum ferritin levels, which reflect a deficiency in iron levels of the body. Reference values for serum ferritin levels in patients with iron deficiency vary. The guidelines generally state a cut-off value of <15–30 ng/mL in apparently healthy individuals. Because ferritin is an acute-phase reactant, higher thresholds are recommended in the presence of infection or inflammation (e.g., <70 ng/mL in adults), and cut-offs may vary by clinical context and guideline [14,15].

Other indicators of iron deficiency include a transferrin saturation (TSAT) of <20%, which reflects iron availability. TSAT is a measure of the proportion of iron-binding sites on transferrin that are occupied by iron; it is calculated as the ratio of serum iron to total iron-binding capacity (TIBC) [16,17]:

$$TSAT(\%) = (Serum\ Iron\ Concentration\ /\ TIBC) \times 100$$

A TSAT level <20% suggests iron deficiency, indicating that there is insufficient iron bound to transferrin, leading to inadequate iron supply to the bone marrow and other tissues [18].

Ferritin levels alone may not accurately reflect iron levels in the presence of chronic diseases or inflammation. Ferritin can be elevated in inflammatory conditions, masking an underlying iron deficiency [12]. The use of TSAT to diagnose iron deficiency is particularly valuable in cases of inflammation [12]. TSAT allows the assessment of iron availability for erythropoiesis and other iron-dependent processes, even when ferritin levels are elevated due to inflammation [19]. Several studies and guidelines recommend using both TSAT and ferritin to diagnose iron deficiency in chronic inflammatory conditions; a recent study proposed TSAT- and ferritin-based indices to support the interpretation of iron status in the context of inflammation [20,21].

Beyond gynecology, iron deficiency has clinically relevant effects on the kidney–cardiovascular axis. In chronic kidney disease and heart failure, functional iron deficiency is common and associated with poorer exercise capacity, quality of life (QoL), and outcomes even when Hb levels are preserved [22,23]. Because ferritin behaves as an acute-phase reactant in these inflammatory states [22], contemporary practice combines TSAT and ferritin findings to define iron deficiency [23,24]. TSAT is a useful indicator in inflammation, as it reflects bioavailable iron for erythropoiesis when ferritin may be spuriously elevated [22,24].

Although functional iron deficiency is associated with various diseases, it has not been extensively studied in benign gynecological conditions, including uterine adenomyosis. Therefore, the present study aimed to assess the frequency of functional iron deficiency in patients with benign gynecological diseases.

## Materials and methods

### Study design and population

We conducted a retrospective cross-sectional observational study using electronic medical records from the first outpatient visit. Specifically, we performed a retrospective analysis of electronic medical records of patients who presented for their initial consultation at our hospital between April 2019 and January 2025. At our institution, a complete blood count is routinely performed as part of the initial assessment of new patients. Further iron studies, including serum iron, TIBC, and particularly serum ferritin, were performed at the discretion of the attending physician based on the clinical context.

Non-menopausal women aged 30–55 years who underwent initial blood testing were included. The exclusion criteria were as follows: (1) pregnancy or the postpartum period; (2) malignant tumors; (3) postmenopausal status; (4) receipt of iron therapy prior to blood sampling; (5) autoimmune diseases; (6) renal diseases; and (7) insufficient clinical data for analysis.

All analyses were based on this eligible first-visit cohort. For specific analyses involving serum ferritin, the dataset was restricted to data on patients for whom a ferritin value was available from their first visit (Fig 1).

To mitigate selection bias arising from non-protocolized ordering of iron studies, we enrolled all consecutive new patients who met the eligibility criteria, used only first-visit laboratory results, and applied only prespecified, non-discretionary exclusion criteria.

### Study procedures

Clinical data, including the presence of adenomyosis, uterine fibroids, menorrhagia, fatigue symptoms, Hb concentration, TSAT levels, and serum ferritin levels, were extracted from medical records. Uterine fibroids and adenomyosis were

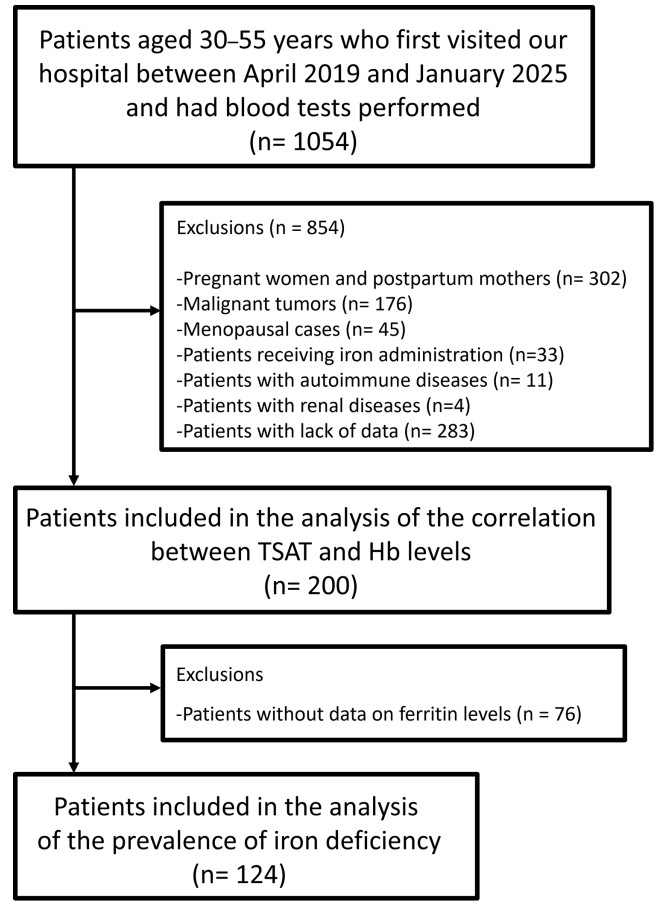

**Fig 1. Study flowchart.**

diagnosed using ultrasonography and magnetic resonance imaging. The participants were categorized into three groups: 1) those with adenomyosis (adenomyosis group); 2) those with uterine myoma (myoma group); and 3) those without structural uterine abnormalities (no uterine structural abnormality group).

## Definitions

We defined anemia as Hb < 11 g/dL to capture clinically significant (moderate-to-severe) anemia. The WHO definition of anemia in non-pregnant women is Hb < 12 g/Dl; however, Hb < 11 g/dL corresponds to at least moderate anemia in severity classification, and it is commonly used in Japanese gynecologic practice for non-pregnant women with gynecologic conditions [25,26].

Absolute iron deficiency was defined as serum ferritin <20 ng/mL. Functional iron deficiency (FID) was defined as TSAT <20%, with ferritin ≥20 ng/mL. Overall iron deficiency was defined as ferritin < 20 ng/mL and/or TSAT < 20%.

## Statistical analysis

Continuous variables were summarized as medians (range) and compared across groups using the Kruskal–Wallis test. Categorical variables were summarized as n (%) and analyzed using the chi-squared or Fisher's exact test, as appropriate. The McNemar's test was used for paired binary comparisons within the same participants.

Agreement between ferritin-based and TSAT-based classifications (ferritin <20 ng/mL vs TSAT <20%) was evaluated using Cohen's kappa, with PABAK (2Po − 1) reported as a sensitivity analysis. Factors associated with discordant ("TSAT-only") classification suggestive of functional iron deficiency (TSAT <20% with ferritin ≥20 ng/mL) were assessed using multivariable logistic regression, and model discrimination was summarized via ROC/AUC (ROC shown in S1 Fig).

Analyses requiring ferritin were restricted to patients with ferritin measured at the first visit. All tests were two-sided, with α = 0.05. Because transferrin TSAT level showed a right-skewed distribution, the primary association between Hb and TSAT was evaluated via Pearson's correlation using log-transformed TSAT; Spearman's rank correlation on untransformed TSAT data was performed as a sensitivity analysis. Coefficients were reported as r for Pearson and ρ for Spearman. Data were analyzed using EZR software and GraphPad Prism 10.

### Ethical approval

This study was reviewed and approved by the Ethics Review Board (approval number: ERB-C-2981; approval date: 10/31/2023) of our institution. As per our institutional policy, patients were informed at the initial visit that their anonymized clinical data may be used for future research purposes. In this study, we included data exclusively from patients who had provided written broad consent for such secondary use. This consent procedure was fully approved by the Ethics Review Board as the ethical basis for this research. This study was conducted in accordance with the principles of the Declaration of Helsinki.

## Results

### Clinical characteristics of patients

During the study period, 1054 patients first visited the hospital, and their blood samples were collected. Based on the study criteria, 854 and 200 patients were initially excluded and included, respectively. The participants were initially categorized into the adenomyosis (42 patients), myoma (108 patients), and no uterine structural abnormality (50 patients) groups. We used two prespecified analysis populations: a correlation cohort (Hb and TSAT levels available at the first visit; summarized in Table 1) and a ferritin analytic cohort (first-visit ferritin levels available) for iron-deficiency frequencies.

After further exclusion of patients without ferritin level measurements, 29 patients were ultimately included in the adenomyosis group, 55 in the myoma group, and 40 in the no uterine structural abnormality group. Patient demographics of the correlation cohort are presented in Table 1. There were no significant differences in age, body mass index, or the frequency of self-reported hypermenorrhea between the three groups. However, a higher proportion of patients in the adenomyosis group reported fatigue compared to the other groups.

**Table 1. Baseline characteristics of the correlation cohort (patients with hemoglobin and transferrin saturation data available at the first visit).**

| | Adenomyosis group (n = 42) | Uterine myoma group (n = 108) | No uterine structural abnormality group (n = 50) | P values |
|---|---|---|---|---|
| Age | 45 [33–52] | 45 [30–54] | 41.5 [30–53] | 0.214 |
| Body mass index | 22.7 [17.5–30.6] | 22.1 [15.7–42.9] | 21.0 [16.1–35.1] | 0.111 |
| Presence of hypermenorrhea | 21 (50.0%) | 60 (55.6%) | 23 (46.0%) | 0.509 |
| Presence of fatigue | 17 (40.4%) | 11 (10.2%) | 4 (8.0%) | < 0.01 |

Median [range]; n (%)

## Frequency of anemia and iron deficiency in the groups differed according to different markers

The frequency of anemia and iron deficiency in each group, as diagnosed based on Hb for anemia and serum ferritin and TSAT levels for iron deficiency, is shown in Fig 2. Anemia was defined as Hb < 11 g/dL. Iron deficiency frequencies were summarized using three marker-based criteria: ferritin-only (serum ferritin <20 ng/mL), TSAT-only (transferrin saturation <20%), and a combined criterion (ferritin <20 ng/mL and/or TSAT <20%).

Iron deficiency was more prevalent than anemia across all groups. Notably, in the adenomyosis group, iron deficiency diagnosed using ferritin and TSAT levels was significantly higher than that diagnosed using ferritin level alone.

## Agreement between ferritin and TSAT based classifications and factors associated with TSAT only discordance

Agreement between ferritin-based and TSAT-based classifications was modest. Among patients with ferritin measured at the first visit (n = 124), 45 (36.3%) met both criteria (ferritin <20 ng/mL and TSAT <20%), 40 (32.3%) met none of the criteria, 14 (11.3%) met the ferritin-only criterion, and 25 (20.2%) met the TSAT-only criterion (TSAT <20% with ferritin ≥20 ng/mL). The agreement between ferritin-based and TSAT-based classifications was fair (Cohen's kappa = 0.375; PABAK = 0.371).

In multivariable logistic regression for TSAT-only classification (TSAT <20% with ferritin ≥20 ng/mL), adjusting for age and hypermenorrhea, adenomyosis showed higher odds compared with no uterine structural abnormality (OR 3.123, 95% CI 0.942–11.30; p = 0.0688), whereas myoma was not associated (OR 1.234, 95% CI 0.399–4.079; p = 0.7184). The model showed modest discrimination (AUC = 0.678, 95% CI 0.564–0.792) (S1 Fig).

## Correlation between TSAT and Hb levels was weak in the adenomyosis group

The correlation between log(TSAT) and Hb concentration in each group is shown in Fig 3. Only a weak correlation trend was observed in the adenomyosis group (r = 0.2123, p = 0.1771), whereas a relatively strong and statistically significant correlation was found in the myoma (r = 0.5465, p < 0.0001) and no uterine structural abnormality groups (r = 0.6945, p < 0.0001).

Using Spearman's rank correlation on untransformed TSAT, we observed the same qualitative pattern: adenomyosis group ρ = 0.358, p = 0.020 (n = 42); myoma group ρ = 0.576, p < 0.0001 (n = 108); no uterine structural abnormality group

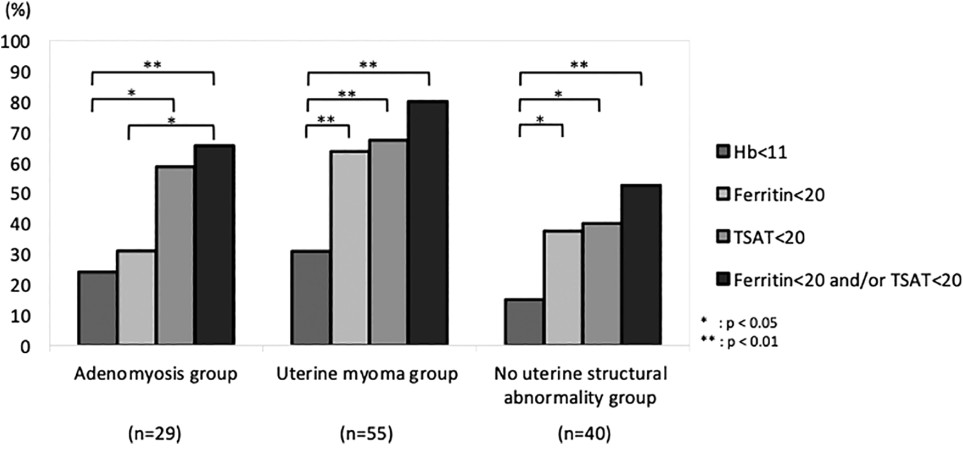

**Fig 2. Frequency of anemia and iron deficiency, as diagnosed using Hb, serum ferritin, and TSAT levels.** In the adenomyosis group, iron deficiency diagnosed using ferritin, and TSAT was significantly higher than that diagnosed using ferritin alone. Hb, hemoglobin; TSAT, transferrin saturation.

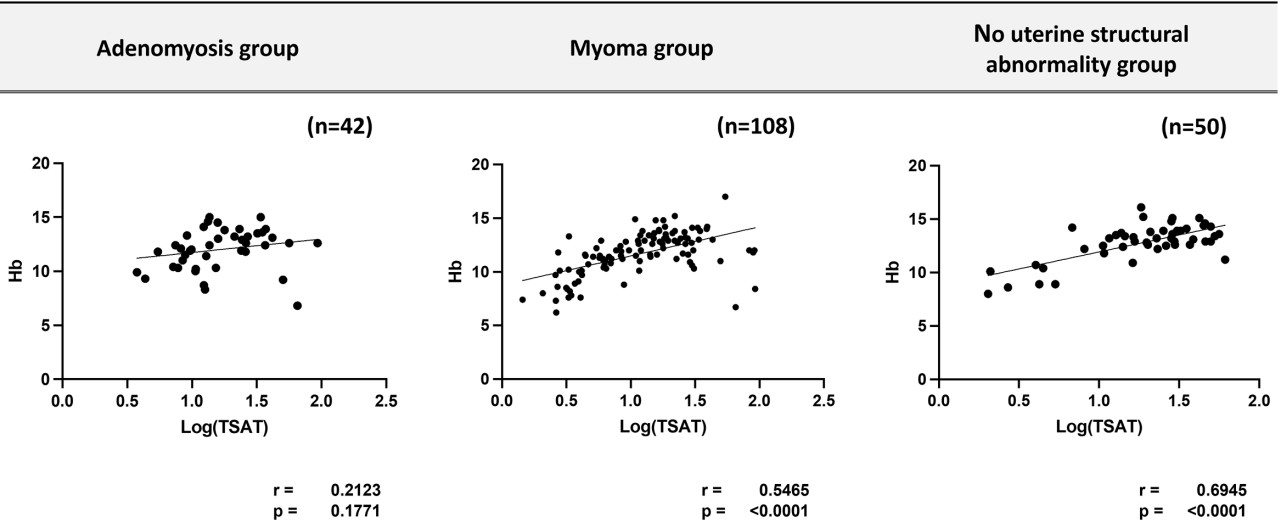

**Fig 3. Correlation between log(TSAT) and Hb concentrations.** The correlation was weak in the adenomyosis group, compared to strong correlations in the myoma and no uterine structural abnormality groups. Hb, hemoglobin; TSAT, transferrin saturation.

$\rho = 0.484$, p = 0.0004 (n = 50). Stratified analyses similarly mirrored Pearson analysis results (adenomyosis with hypermenorrhea: $\rho = 0.218$, p = 0.342, n = 21; without hypermenorrhea: $\rho = 0.563$, p = 0.0079, n = 21).

### Hypermenorrhea may affect the correlation between TSAT and Hb level

Furthermore, the patients were subdivided into six groups based on the presence or absence of hypermenorrhea. Approximately half of patients in each group experienced hypermenorrhea. The correlation between log(TSAT) and Hb levels was examined in these six subgroups. A notable absence of correlation was observed in the adenomyosis group with hypermenorrhea (r = 0.0526, p = 0.8209) (Fig 4).

### Discussion

In the present study, correlation analysis revealed a strong and statistically significant correlation between Hb levels and log(TSAT) in the myoma (r = 0.5465, p < 0.0001) and no uterine structural abnormality groups (r = 0.6945, p < 0.0001). However, only a weak correlation trend was observed in the adenomyosis group (r = 0.2123, p = 0.1771). These findings suggest that patients with adenomyosis may have a higher risk of undiagnosed iron deficiency despite the high frequency of fatigue and other related symptoms in this patient population. The results of this study, which showed a discrepancy between Hb levels and log(TSAT) in patients with uterine adenomyosis as a benign nonmalignant disease, have not been reported previously.

Our study demonstrated that iron deficiency was more prevalent than anemia in all groups. Hypermenorrhea causes iron deficiency, which is generally believed to be partly caused by morphological uterine changes [27,28]. Uterine myoma and adenomyosis are considered similar diseases because their morphological changes cause excessive menstruation. However, in the adenomyosis group, the frequency of iron deficiency diagnosed using serum ferritin and TSAT levels was significantly higher than that diagnosed using ferritin level alone. In contrast, the myoma and no uterine structural abnormality groups showed similar frequencies of iron deficiency, regardless of the diagnostic criteria. In the uterine myoma and no uterine structural abnormality groups, the frequency of anemia (defined as Hb < 11 g/dL) was 30.9% and 15.0%, respectively. The frequency of iron deficiency (defined as serum ferritin <20 ng/mL) in these groups was 63.6% and

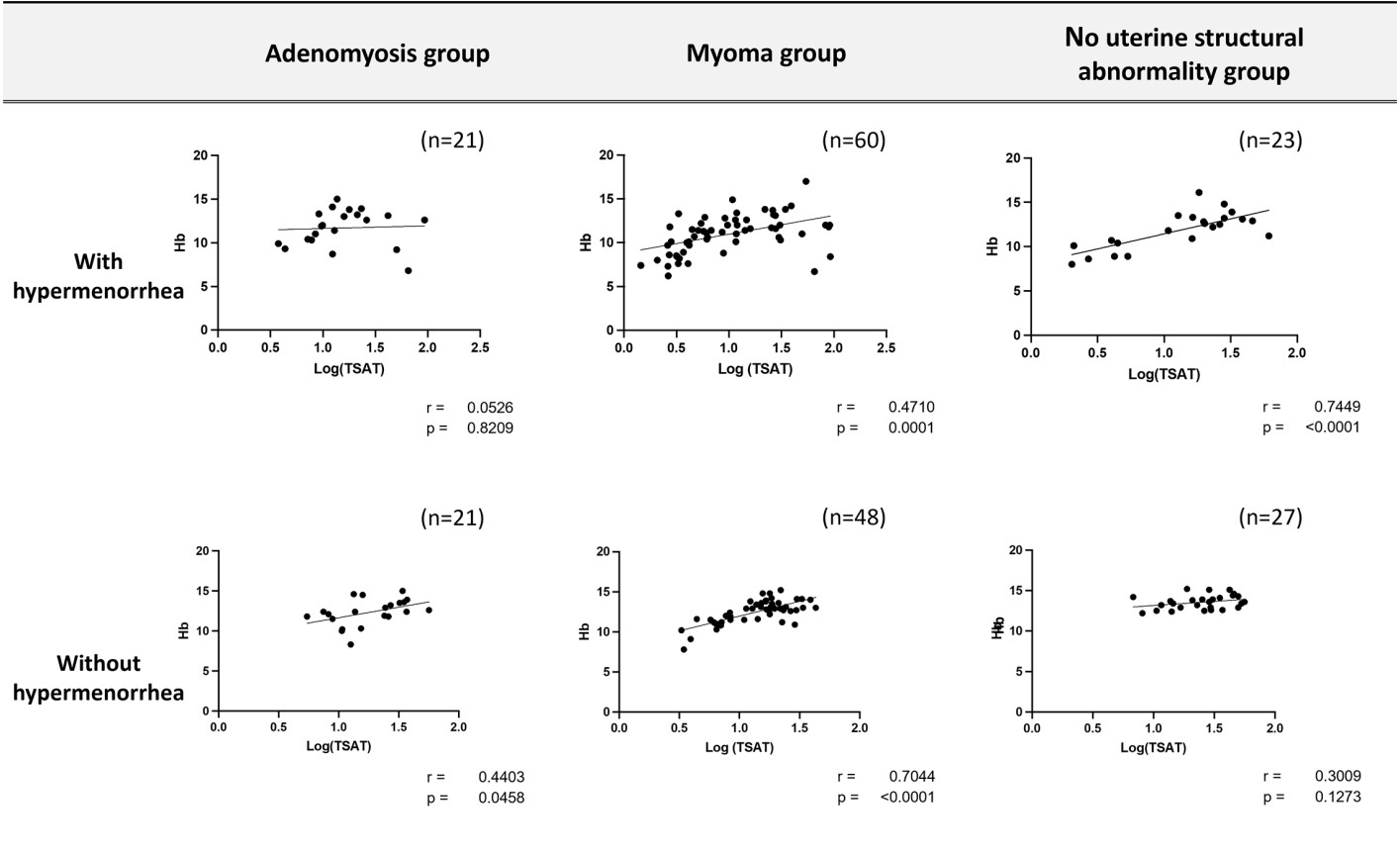

**Fig 4. Correlation between log(TSAT) and Hb concentrations.**A notable absence of correlation was observed only in the adenomyosis group with hypermenorrhea (r = 0.0526, p = 0.8209). Hb, hemoglobin; TSAT, transferrin saturation.

37.5%, respectively. These findings indicate that iron deficiency was more common than anemia in these patient populations, highlighting the importance of assessing iron status with markers such as ferritin and Hb concentration. In contrast, in the adenomyosis group, the frequency of iron deficiency diagnosed using ferritin alone was 31.0%, similar to that of anemia in this group. However, when iron deficiency was defined as ferritin <20 ng/mL and/or TSAT <20%, the frequency in the adenomyosis group was significantly higher (65.5%).

TSAT reflects iron availability for erythropoiesis and other iron-dependent processes; it is calculated as the ratio of serum iron concentration to TIBC, and it is a valuable tool for assessing iron status. For chronic inflammatory diseases, such as cancer, heart failure, inflammatory bowel disease, and chronic kidney disease, the literature suggests that incorporating TSAT and serum ferritin into diagnostic approaches enhances iron deficiency screening [29]. The frequency of iron deficiency in young women exceeds that of anemia [30]. Similarly, our study found that this trend persisted among premenopausal patients with benign gynecological conditions. TSAT has been analyzed in various diseases and is increasingly used in clinical practice. However, its utility in assessing iron deficiency in adenomyosis has not been elucidated. Our results suggest that TSAT may be a more sensitive indicator of iron deficiency in adenomyosis than ferritin level alone.

To assess whether ferritin- and TSAT-based criteria identify the same patients, we evaluated the concordance between ferritin <20 ng/mL and TSAT <20% among those with ferritin measured at the first visit (n = 124). The agreement was fair (Cohen's κ = 0.375; PABAK = 0.371), indicating only partial overlap. Notably, 20.2% of the patients fulfilled a TSAT-only classification (TSAT <20% with ferritin ≥20 ng/mL), a pattern consistent with functional iron deficiency that ferritin alone may miss in inflammatory states. In multivariable logistic regression adjusted for age and hypermenorrhea, adenomyosis showed a trend toward higher odds of TSAT-only classification versus no structural abnormality (OR 3.123, 95% CI 0.942–11.30; p = 0.0688), whereas myoma was not associated (OR 1.234, 95% CI 0.399–4.079; p = 0.7184). The model's discrimination was modest (AUC = 0.678). The AUC reflects within-sample discrimination in this cohort.

Patients with iron deficiency may present with symptoms such as fatigue, without a decrease in Hb levels, and symptoms may improve with the administration of iron supplements. For example, in patients with heart failure, functional or absolute iron deficiency independent of anemia is associated with decreased exercise ability, reduced QoL, and a poor prognosis [2]. Intravenous iron therapy for such patients may improve exercise capacity, the New York Heart Association class, and QoL [2]. In women with functional or absolute iron deficiency without anemia, the administration of iron improves physical function and decreases fatigue scores [8,9,31,32]. Furthermore, in patients with adenomyosis, treatment for iron deficiency may improve symptoms.

Non-anemic iron deficiency is clinically important because iron supports oxidative metabolism, mitochondrial/myocyte function, and neurotransmitter synthesis; therefore fatigue, cognitive complaints, headaches, and reduced exercise tolerance can occur even when Hb level is normal [12,13]. In inflammatory benign gynecologic conditions, ferritin may be elevated as an acute-phase reactant; consequently, TSAT helps reveal iron-restricted erythropoiesis when ferritin and hemoglobin appear preserved [19]. In our cohort, the weaker TSAT–Hb correlation observed in adenomyosis suggests that TSAT may aid the evaluation of symptomatic patients with normal Hb levels, particularly when inflammation is suspected.

Mechanistically, this inflammatory milieu likely mediates iron restriction via the hepcidin–ferroportin axis. Adenomyosis displays macrophage infiltration, NF-κB activation, and increased interleukin-6 (IL-6) expression by endometrial stromal cells [33]. IL-6 induces hepatic hepcidin via JAK/STAT3, and hepcidin internalizes the iron exporter ferroportin on enterocytes and macrophages, thereby reducing intestinal iron absorption and reticuloendothelial iron release [34]. The net effect is iron sequestration with low circulating iron and low TSAT, despite normal or increased ferritin, which is the biochemical signature of functional iron deficiency. Analogous hepcidin-related iron sequestration has been reported in endometriosis (elevated hepcidin in peritoneal fluid and menstrual blood and increased iron storage in peritoneal macrophages), suggesting a shared inflammation–hepcidin pathway that may be involved in adenomyosis [35,36]. This mechanism aligns with our observation that TSAT was frequently reduced while hemoglobin and ferritin were not proportionally decreased.

IL-6–hepcidin-mediated inflammation in adenomyosis restricts circulating iron, lowering TSAT despite preserved ferritin; accordingly, Hb/ferritin-only assessments may overlook functional iron deficiency. Differences from previous studies may be due to varying methodologies: we incorporated TSAT and explicitly evaluated Hb–TSAT coupling, whereas studies relying on Hb and/or ferritin alone were less sensitive to functional iron deficiency in inflammatory states.

Although a strong and statistically significant correlation between log(TSAT) and Hb levels was observed in the uterine myoma and no uterine structural abnormality groups, the markedly weaker correlation pattern between log(TSAT) and Hb levels in the adenomyosis group, particularly in patients with excessive menstruation, suggests that the etiology of iron deficiency in adenomyosis may be beyond structural changes and acute bleeding. Adenomyosis is characterized by the presence of ectopic endometrial tissue within the myometrium, leading to chronic inflammation and altered immune response [37]. A chronic inflammatory state may contribute to the development of functional iron deficiency, in which iron is sequestered and not readily available for erythropoiesis, resulting in low TSAT levels even when Hb and ferritin levels are maintained. Local inflammatory mediators such as IL-6 and IL-8 are elevated in adenomyosis, and an increased

number of inflammatory cells such as macrophages and natural killer cells are present in the endometrium and myometrium [38–40]. These findings support the hypothesis that adenomyosis is associated with a chronic inflammatory state contributing to functional iron deficiency.

Furthermore, our study found that this trend persisted among premenopausal patients with benign gynecological conditions. In the adenomyosis group, the frequency of iron deficiency diagnosed using serum ferritin and TSAT levels was higher than the frequency of iron deficiency diagnosed using serum ferritin level alone. Furthermore, inflammatory states elevate hepcidin levels, thereby diminishing iron absorption and promoting iron sequestration, leading to functional iron deficiency without serum ferritin reduction [41]. Hence, relying solely on serum ferritin levels may lead to underdiagnosis. In adenomyosis, the inclusion of TSAT in the diagnostic criteria underscores the chronic inflammatory underpinnings potentially inherent to the condition, as reflected by the frequency of augmented iron deficiency.

In addition, the adenomyosis group with hypermenorrhea exhibited a negligible correlation between log(TSAT) and Hb levels, unlike the myoma group, where a strong correlation was evident. This disparity indicates that the etiology of iron deficiency in adenomyosis may extend beyond structural changes. The interplay between acute blood loss from excessive menstrual bleeding and a chronic inflammatory state may disrupt iron metabolic profile, resulting in low TSAT levels, even when the Hb level is maintained. Notably, the patient may have iron deficiency with low TSAT levels even when Hb is maintained.

Although endometriosis, which is similar to adenomyosis, was not explored in this study, it may share a similar pathophysiology involving systemic chronic inflammation, thus warranting further investigation of potential therapeutic interventions.

This study had several limitations, primarily due to its retrospective design. Although a complete blood count was routinely performed, the decision to measure serum ferritin was not uniformly protocolized and was determined by the attending physician. This approach may have introduced a potential source of selection bias. We speculate that physicians were more inclined to order comprehensive iron studies for patients with suspected chronic inflammatory conditions such as adenomyosis, while attributing anemia primarily to blood loss in patients with uterine myomas. This practice may explain the higher number of exclusions observed in the myoma group. In addition to this primary limitation, other factors must be considered. These include the relatively small patient cohort, particularly in the adenomyosis group; the non-synchronization of blood sampling with the menstrual cycle; and the omission of hormonal treatment history, all of which could affect iron metabolism. These factors underscore the need for future prospective observational studies.

## Conclusions

Our findings indicate that TSAT measurement in patients with adenomyosis is a critical diagnostic parameter for iron deficiency and could potentially inform diagnostic and management strategies. The incorporation of TSAT into the diagnostic repertoire enhances the detection of iron deficiency, particularly functional iron deficiency, which may not be identified using serum ferritin levels alone. Therefore, the assessment of TSAT may offer a more comprehensive evaluation of iron level in patients with adenomyosis, leading to improved therapeutic outcomes. However, further research is warranted to confirm these findings and explore the optimal incorporation of TSAT measurements in the clinical management of iron deficiency in uterine adenomyosis.

## Supporting information

**S1 Fig. ROC curve for predicting functional iron deficiency classification.** ROC curve of the multivariable logistic regression model for TSAT-only classification (TSAT <20% with ferritin ≥20 ng/mL) among patients with ferritin measured at the first visit (n = 124). Predictors included diagnosis group (adenomyosis and myoma, with no uterine structural abnormality as the reference), age (years), and hypermenorrhea. The area under the ROC curve (AUC) was 0.678 (95% CI 0.563–0.793).
(PDF)

## Author contributions

**Conceptualization:** Mari Kawamata, Fumitake Ito, Taisuke Mori.

**Data curation:** Mari Kawamata, Fumitake Ito, Yuko Izumi, Akihisa Katayama.

**Formal analysis:** Mari Kawamata, Fumitake Ito, Nanami Tahara, Kazuya Yabumoto, Yuko Izumi, Akihisa Katayama.

**Funding acquisition:** Fumitake Ito.

**Investigation:** Mari Kawamata, Fumitake Ito, Nanami Tahara, Akiyo Kakibuchi, Kazuya Yabumoto.

**Methodology:** Mari Kawamata, Fumitake Ito.

**Project administration:** Taisuke Mori.

**Supervision:** Taisuke Mori.

**Validation:** Fumitake Ito.

**Writing – original draft:** Mari Kawamata, Fumitake Ito, Taisuke Mori.

**Writing – review & editing:** Mari Kawamata, Fumitake Ito, Nanami Tahara, Kazuya Yabumoto, Yuko Izumi, Akihisa Katayama, Taisuke Mori.

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
