## [Decision Letter · Decision Letter 0]

13 Aug 2025

PONE-D-25-29781
Lack of correlation between transferrin saturation and anemia-related biomarkers in patients with uterine adenomyosis
PLOS ONE

Dear Dr. Ito,

Thank you for submitting your manuscript to PLOS ONE. After careful consideration, we feel that it has merit but does not fully meet PLOS ONE’s publication criteria as it currently stands. Therefore, we invite you to submit a revised version of the manuscript that addresses the points raised during the review process.

First, align the title, the aim stated in the Abstract, and the aim in the Introduction. If no intervention was analyzed, remove references to “therapeutic iron repletion.” Throughout, use frequency rather than prevalence. Define anemia and iron deficiency explicitly—distinguishing absolute from functional iron deficiency—and state clearly how ferritin and TSAT were used (individually and in combination). Please revise the passages that might imply ferritin/TSAT diagnose anemia; anemia should be defined by hemoglobin, whereas ferritin/TSAT define iron deficiency.

Second, clarify study design and cohort construction. Specify how the study population was selected and whether TSAT, ferritin, serum iron, and TIBC were ordered routinely or based on presenting complaints. Reconcile the retrospective timeframe (2019–2025) with IRB approval and consent procedures, and define the target population and any clinician-applied exclusion criteria, explaining how selection bias was minimized. Please also explain why missing ferritin disproportionately excluded patients in the myoma group compared with adenomyosis and controls.

Third, address clinical interpretation. Analyze whether symptom severity correlates with TSAT within adenomyosis and myoma (e.g., PBAC/menorrhagia and dysmenorrhea scores or appropriate binary proxies) and report effect sizes. Expand the Discussion on iron deficiency without anemia, and briefly comment on renal and cardiovascular consequences of iron deficiency and their relevance to interpreting TSAT in inflammatory states. Strengthen the mechanistic link among adenomyosis-related inflammation, hepcidin, and functional iron deficiency, and moderate language that overstates a “lack of correlation.”

We look forward to receiving your revised manuscript.

Kind regards,

Kazunori Nagasaka

Academic Editor

PLOS ONE

Journal Requirements:

“This work was partially supported by Grants-in-Aid for Scientific Research 21K09546 from the Ministry of Education, Culture, Sports, Science, and Technology (Japan).”

“The authors have nothing to disclose.”

4. We note that your Data Availability Statement is currently as follows: All relevant data are within the manuscript and in Supporting Information files.

Additional Editor Comments:

Dear Authors,

Thank you for submitting “Lack of correlation between transferrin saturation and anemia-related biomarkers in patients with uterine adenomyosis.”

We and both external reviewers found the clinical question important.

However, substantial clarifications and additional analyses are required before the manuscript can be considered further.

Please provide a detailed, point-by-point response to each reviewer; Reviewer 2 raises particularly important issues that will strengthen your manuscript.

First, align the title, the aim stated in the Abstract, and the aim in the Introduction. If no intervention was analyzed, remove references to “therapeutic iron repletion.” Throughout, use frequency rather than prevalence. Define anemia and iron deficiency explicitly—distinguishing absolute from functional iron deficiency—and state clearly how ferritin and TSAT were used (individually and in combination).

Please revise the passages that might imply ferritin/TSAT diagnose anemia; anemia should be defined by hemoglobin, whereas ferritin/TSAT define iron deficiency.

Second, clarify study design and cohort construction. Specify how the study population was selected and whether TSAT, ferritin, serum iron, and TIBC were ordered routinely or based on presenting complaints. Reconcile the retrospective timeframe (2019–2025) with IRB approval and consent procedures, and define the target population and any clinician-applied exclusion criteria, explaining how selection bias was minimized.

Please also explain why missing ferritin disproportionately excluded patients in the myoma group compared with adenomyosis and controls.

Third, address clinical interpretation. Analyze whether symptom severity correlates with TSAT within adenomyosis and myoma (e.g., PBAC/menorrhagia and dysmenorrhea scores or appropriate binary proxies) and report effect sizes. Expand the Discussion on iron deficiency without anemia, and briefly comment on renal and cardiovascular consequences of iron deficiency and their relevance to interpreting TSAT in inflammatory states. Strengthen the mechanistic link among adenomyosis-related inflammation, hepcidin, and functional iron deficiency, and moderate language that overstates a “lack of correlation.”

Please submit a clean and a tracked-changes manuscript, updated figures/tables, and a comprehensive response to reviewers.

If any of the requested clarifications or analyses cannot be fully addressed, please provide a clear rationale in your rebuttal letter (e.g., data availability, ethical constraints, or methodological limitations) and reflect this in the revised manuscript’s limitations.

We look forward to your revision.

Plos One

Kazunori Nagasaka

Reviewers' comments:

Reviewer's Responses to Questions

**Comments to the Author**

1. Is the manuscript technically sound, and do the data support the conclusions?

Reviewer #1: Yes

Reviewer #2: Partly

2. Has the statistical analysis been performed appropriately and rigorously? 

Reviewer #1: Yes

Reviewer #2: I Don't Know

3. Have the authors made all data underlying the findings in their manuscript fully available?

Reviewer #1: Yes

Reviewer #2: Yes

4. Is the manuscript presented in an intelligible fashion and written in standard English?

Reviewer #1: Yes

Reviewer #2: Yes

5. Review Comments to the Author

Reviewer #1: Kindly use word Frequency instead of Prevalence. Should deliberate more in discussion regarding iron deficiency without anemia, as this is much prevalent. Should also comment on kidney and cardiovascular dearragements that occur in iron deficiency.

Reviewer #2: This is a well thought out research. However, there are some areas that will need clarification, reconstruction and adjustment in order to present a logical conclusion some of which are highlighted below as questions, observations and suggestions.

There is disconnection between the title, the aim as written in the abstract (line 27-28) and the study aim written in the introduction section (line 94-96)

How was study population selected and what role did their presenting complain play in the selection?

what is therapeutic iron repletion?

How is functional iron deficiency diagnosed especially in this study population?

If this was a retrospective review, how was informed consent gotten from patients prior to participation in the study? What exactly is the study design used? it is noted also that ethical approval was gotten on 31/10/2023 but the study was a retrospective review from 01/04/2019 to 31/01/2025. The study design, participants recruitment is ambiguous. it needs to be better written. Who are the target population for this study? What factors were considered by the attending obstetrician-gynecologist to deem a patient unfit for participating in the study? How will that not introducing selection bias?

Is it a routine practice to test for TSAT and serum ferritin levels? Was serum iron also assayed for

There were more patients excluded in the myoma group (49%) compared with Adenomyosis group (31%) and those without structural anomalies (20%) because of lack of ferritin assay. could you provide some explanation to this?

Table 1 included all the initial patients despite that some were excluded from the study because of lack of ferritin assay.

Line 157 -163 needs to be written to ensure clarity. How is iron deficiency diagnosed using ferritin and TSAT combined as opposed to using ferritin or TSAT alone.

what was the justification for the log transformation of TSAT? Did you consider a sensitivity analysis without the log transformation? Assuming the reason for the transformation was the non-normality of the data, did you consider doing a different form of correlation?

The pathophysiology of adenomyosis includes some chronic inflammatory process. what is the effect of that on the TSAT level as compared to the other groups? What explanation can you give as to why the findings in this study was different from previous studies? (line 199-201)

Also since all the patients did not present with hypermenorrhea, were there other factors considered (as cofounders) that could have been responsible for the iron deficiency or anemia?

line 209-214, This should be rewritten so as not to give the impression of comparing prevalence of anemia diagnosed by Hb concentration and ferritin. and since it is common knowledge that there is a subtype of Iron deficiency that do not present with anemia, the Hb levels cannot be used to diagnose iron deficiency and likewise ferritin and TSAT should not be used to diagnose anemia.

6. PLOS authors have the option to publish the peer review history of their article (what does this mean?). If published, this will include your full peer review and any attached files.

Reviewer #1: **Yes:**Bader Faiyaz Zuberi

Reviewer #2: No

---

## [Author Response · Author response to Decision Letter 1]

30 Sep 2025

Additional Editor Comments:

#1

First, align the title, the aim stated in the Abstract, and the aim in the Introduction. If no intervention was analyzed, remove references to “therapeutic iron repletion.”

Response:

Thank you for pointing this out. We removed references to “therapeutic iron repletion.”

#2

Throughout, use frequency rather than prevalence.

Response:

We agree with the reviewer and have replaced “prevalence” with “frequency” throughout the manuscript, including headings and figure text.

#3

Define anemia and iron deficiency explicitly—distinguishing absolute from functional iron deficiency—and state clearly how ferritin and TSAT were used (individually and in combination).

Response:

We appreciate the suggestion. We added a Definitions subsection specifying the following: anemia = Hb <11 g/dL; absolute iron deficiency = ferritin <20 ng/mL; functional iron deficiency = TSAT <20%, with ferritin ≥20 ng/mL; overall iron deficiency = ferritin <20 ng/mL and/or TSAT <20%.

#4

Please revise the passages that might imply ferritin/TSAT diagnose anemia; anemia should be defined by hemoglobin, whereas ferritin/TSAT define iron deficiency.

Response:

We completely agree with your comment and reworded the Abstract, Results, and Discussion to avoid suggesting that ferritin/TSAT levels can be used to diagnose anemia. We have stated explicitly that anemia was defined using Hb only, whereas iron deficiency was defined using ferritin and/or TSAT.

#5

Second, clarify study design and cohort construction. Specify how the study population was selected and whether TSAT, ferritin, serum iron, and TIBC were ordered routinely or based on presenting complaints.

Response:

We appreciate the editor’s suggestion. We clarified that the study is a retrospective observational study of consecutive first-visit gynecology outpatients (Apr 2019–Jan 2025). CBC is routine at the first visit, whereas iron studies (serum iron, TIBC, ferritin)—from which TSAT is calculated—were ordered at the attending physician’s discretion based on clinical context. These clarifications were made by revising Study design and population.

#6

Reconcile the retrospective timeframe (2019–2025) with IRB approval and consent procedures.

Response:

We appreciate the suggestion. The study was conducted under our institution’s broad written consent program. The IRB approved the retrospective use of records from April 2019 to January 2025. Participants had already provided broad written consent permitting the use of de-identified clinical data for research.

#7

Define the target population and any clinician-applied exclusion criteria, explaining how selection bias was minimized.

Response:

Thank you for this helpful comment. We specified the target population as non-menopausal women aged 30–55 years who underwent first-visit blood testing. We listed prespecified exclusions. To minimize selection bias, we used consecutive sampling, applied a uniform first-visit laboratory policy, and limited exclusions to the above a priori criteria. These changes were made by revising Materials and Methods → Study design and population.

#8

Please also explain why missing ferritin disproportionately excluded patients in the myoma group compared with adenomyosis and controls.

Response:

Thank you for pointing this out. In Discussion (Limitations), we added that iron studies were not protocolized and were ordered at clinician discretion.

#9

Third, address clinical interpretation. Analyze whether symptom severity correlates with TSAT within adenomyosis and myoma (e.g., PBAC/menorrhagia and dysmenorrhea scores or appropriate binary proxies) and report effect sizes.

Response:

We appreciate the suggestion. Owing to the retrospective design, validated severity instruments (PBAC, dysmenorrhea VAS) were not systematically collected; only binary proxies (self-reported heavy menstrual bleeding and fatigue) were available. We therefore stratified by heavy menstrual bleeding (Fig. 4), where the Hb–TSAT correlation was absent in adenomyosis, supporting functional iron deficiency despite preserved Hb. We acknowledge this limitation and plan a prospective study collecting standardized severity measures to quantify severity–TSAT associations.

#10

Expand the Discussion on iron deficiency without anemia, and briefly comment on renal and cardiovascular consequences of iron deficiency and their relevance to interpreting TSAT in inflammatory states.

Response:

Thank you for this helpful comment. We expanded the final paragraphs of Introduction and Discussion to summarize iron deficiency without anemia and briefly noted the renal/cardiovascular effects of iron deficiency, emphasizing their relevance to interpreting TSAT under inflammatory conditions.

#11

Strengthen the mechanistic link among adenomyosis-related inflammation, hepcidin, and functional iron deficiency.

Response：

We appreciate the suggestion. We strengthened the mechanistic relationship in Introduction, outlining how adenomyosis-related inflammation can increase hepcidin, leading to functional iron deficiency (i.e., low TSAT levels, despite preserved ferritin levels).

#12

Moderate language that overstates a “lack of correlation.”

Response：

Thank you for this helpful comment. We have moderated the wording and revised the title to avoid implying an absolute lack of correlation.

Review Comments to the Author

Reviewer #1:

#13

Kindly use word Frequency instead of Prevalence.

Response:

We appreciate the suggestion. We agree with the reviewer and have replaced “prevalence” with “frequency” throughout the manuscript, including headings and figure text.

#14

Should also comment on kidney and cardiovascular dearragements that occur in iron deficiency. Should deliberate more in discussion regarding iron deficiency without anemia, as this is much prevalent.

Response:

Thank you for this helpful comment. We expanded the Introduction and Discussion sections to summarize iron deficiency without anemia and briefly noted the effects of iron deficiency on the renal/cardiovascular system, emphasizing their relevance to interpreting TSAT under inflammatory conditions.

Reviewer #2: This is a well thought out research. However, there are some areas that will need clarification, reconstruction and adjustment in order to present a logical conclusion some of which are highlighted below as questions, observations and suggestions.

#15

There is disconnection between the title, the aim as written in the abstract (line 27-28) and the study aim written in the introduction section (line 94-96).

Response:

We appreciate the reviewer’s point. We have harmonized the study aim across the title, Abstract, and Introduction. In Abstract, we removed treatment-leaning language and clarified the diagnostic focus.

#16

How was study population selected and what role did their presenting complain play in the selection?

Response:

We appreciate the opportunity to clarify this point. We clarified that this is a retrospective observational study of consecutive first-visit gynecology outpatients (Apr 2019–Jan 2025). CBC is routine at the first visit, whereas iron studies (serum iron, TIBC, ferritin)—from which TSAT is calculated—were ordered at the attending physician’s discretion based on clinical context. These clarifications were made by revising Study design and population.

#17

What is therapeutic iron repletion?

Response:

Thank you for pointing this out. We removed any reference to “therapeutic iron repletion” in the manuscript.

#18

How is functional iron deficiency diagnosed especially in this study population?

Response:

We appreciate the suggestion. We added a Definitions subsection specifying the following: anemia = Hb <11 g/dL; absolute iron deficiency = ferritin <20 ng/mL; functional iron deficiency = TSAT <20%, with ferritin ≥20 ng/mL; overall iron deficiency = ferritin <20 ng/mL and/or TSAT <20%.

#19

If this was a retrospective review, how was informed consent gotten from patients prior to participation in the study?

Response:

We appreciate the suggestion. The study was conducted under our institution’s broad written consent program. The IRB approved the retrospective use of records from April 2019 to January 2025. Participants had already provided broad written consent permitting the use of de-identified clinical data for research.

#20

What exactly is the study design used? it is noted also that ethical approval was gotten on 31/10/2023 but the study was a retrospective review from 01/04/2019 to 31/01/2025.

Response:

We are grateful for this suggestion. We have stated that the study is a retrospective observational analysis of consecutive first-visit gynecology outpatients. Regarding oversight and consent, we clarified that IRB approval (31 Oct 2023) was obtained under our institution’s broad written consent framework for secondary use of clinical data, which authorized the retrospective review of records from April 2019 to January 2025.

#21

The study design, participants recruitment is ambiguous. it needs to be better written.

Response:

We appreciate the reviewer’s point. We revised Study design and population to clarify the patient population (non-menopausal women aged 30–55 years with first-visit blood testing) included in the study. These clarifications were made by revising Study design and population.

#22

Who are the target population for this study?

Response:

Thank you for this helpful comment. We revised Study design and population to explicitly state the target population: non-menopausal women aged 30–55 years who presented for an initial gynecology visit and underwent first-visit blood testing.

#23

What factors were considered by the attending obstetrician-gynecologist to deem a patient unfit for participating in the study? How will that not introducing selection bias?

Response:

Thank you for highlighting this point. We agree that clinician-discretionary exclusion can introduce selection bias. In our study, no exclusions were made at clinicians’ discretion; only the prespecified, non-discretionary criteria outlined in Study design and population. We have removed expressions that could suggest clinician-discretionary exclusion. Consecutive enrollment was undertaken to minimize selection bias.

#24

Is it a routine practice to test for TSAT and serum ferritin levels? Was serum iron also assayed for

Response:

We appreciate the reviewer’s question. A complete blood count is routinely performed on the first visit. Iron studies (serum iron, TIBC, and ferritin, used to calculate TSAT) are ordered at the attending physician’s discretion according to clinical context. As this information is already specified in Study Design and Population, we did not make further textual changes.

#25

There were more patients excluded in the myoma group (49%) compared with Adenomyosis group (31%) and those without structural anomalies (20%) because of lack of ferritin assay. could you provide some explanation to this?

Response:

Thank you for pointing this out. We added to Discussion (Limitations) that iron studies were not protocolized and were ordered at the clinician’s discretion.

＃26

Table 1 included all the initial patients despite that some were excluded from the study because of lack of ferritin assay.

Response:

We appreciate the opportunity to clarify. Table 1 is now explicitly defined as the correlation cohort (patients with Hb and TSAT levels available at the first visit). In Results, we stated that two prespecified analysis populations were used: a correlation cohort (Hb and TSAT levels available; summarized in Table 1) and a ferritin analytic cohort (first-visit ferritin available) for iron-deficiency frequencies.

#27

Line 157 -163 needs to be written to ensure clarity. How is iron deficiency diagnosed using ferritin and TSAT combined as opposed to using ferritin or TSAT alone.

Response

Thank you for this helpful comment. We agree and rewrote the second sentence of the paragraph in Results titled “Frequency of anemia and iron deficiency in the groups differed according to different markers” to specify the three marker-based criteria and separate anemia.

＃28

What was the justification for the log transformation of TSAT? Did you consider a sensitivity analysis without the log transformation? Assuming the reason for the transformation was the non-normality of the data, did you consider doing a different form of correlation?

Response:

We appreciate the reviewer’s thoughtful suggestion. We clarified in Statistical analysis that TSAT was right-skewed; therefore, we log-transformed TSAT to better satisfy Pearson correlation assumptions. As a sensitivity analysis, Spearman’s rank correlation was performed on untransformed TSAT. We reported in Results that the qualitative group pattern was consistent in Spearman (adenomyosis showing the weakest Hb–TSAT association).

＃29

The pathophysiology of adenomyosis includes some chronic inflammatory process. what is the effect of that on the TSAT level as compared to the other groups?

Response:

Thank you for this helpful question. We strengthened the mechanistic context in Introduction, discussion how adenomyosis-related inflammation can increase hepcidin, leading to functional iron deficiency (i.e., low TSAT despite preserved/normal-to-elevated ferritin). We show that adding TSAT nearly doubles iron-deficiency detection in adenomyosis (31.0-65.5%)１but changes little in the myoma and non-structural control groups. Consistently, the Hb-TSAT correlation is weak in adenomyosis (r=0.21; with hypermenorrhea r=0.05) and strong in myoma (r=0.55) and controls (r=0.69). These comparisons are now clearly presented in the Results (Fig.2-4) and emphasized in the Discussion.

#30

What explanation can you give as to why the findings in this study was different from previous studies? (line 199-201)

Response:

Thank you for the thoughtful suggestion. We clarified this point in Discussion. Immediately after the sentence on mechanistic relationship, we added a sentence explaining the methodological differences, namely, the use of TSAT in addition to ferritin and the explicit evaluation of Hb–TSAT coupling.

＃31

Also since all the patients did not present with hypermenorrhea, were there other factors considered (as cofounders) that could have been responsible for the iron deficiency or anemia?

Response:

We appreciate the point. In this retrospective dataset, standardized measures for potential confounders beyond hypermenorrhea were not systematically available.

＃32

Line 209-214, This should be rewritten so as not to give the impression of comparing prevalence of anemia diagnosed by Hb concentration and ferritin. and since it is common knowledge that there is a subtype of Iron deficiency that do not present with anemia, the Hb levels cannot be used to diagnose iron deficiency and likewise ferritin and TSAT should not be used to diagnose anemia.

Response:

We agree with the reviewer’s comment. We rewrote the second paragraph of Discussion to clarify that anemia was defined using Hb only, whereas iron deficiency was defined using ferritin and/or TSAT, and we reported these as separate frequencies. We deleted expressions that could suggest direct comparison of “anemia diagnosed by ferritin” and instead described marker-based iron-deficiency frequencies alongside Hb-defined anemia.

---

## [Decision Letter · Decision Letter 1]

22 Dec 2025

PONE-D-25-29781R1
Characteristics of transferrin saturation and anemia-related biomarkers in patients with uterine adenomyosis
PLOS One

Dear Dr. Ito,

Thank you for submitting your manuscript to PLOS ONE. After careful consideration, we feel that it has merit but does not fully meet PLOS ONE’s publication criteria as it currently stands. Therefore, we invite you to submit a revised version of the manuscript that addresses the points raised during the review process.

Based on the reviewer comments and my own assessment, major revision is required for the manuscript to meet PLOS ONE’s publication criteria. Revisions required for consideration of acceptance include clarification of the study design, clearer definition of diagnostic criteria with appropriate references, improved description of the statistical methods to ensure reproducibility, and revision of the abstract to ensure that all claims are directly supported by quantitative results. The ethical considerations should also be clearly stated and appropriately positioned in the manuscript. Additional statistical analyses and citation of the specific reference suggested by the reviewer are recommended but not mandatory, provided that the authors justify their chosen approach and ensure that the conclusions are fully supported by the data. Where reviewer comments overlap, authors should prioritize changes that improve methodological clarity and alignment between the data and the conclusions. This decision is based on scientific soundness, transparency, and support of conclusions by the data, in accordance with PLOS ONE’s publication criteria, and not on novelty or perceived impact.

We look forward to receiving your revised manuscript.

Kind regards,

Kazunori Nagasaka

Academic Editor

PLOS One

Journal Requirements:

**Additional Editor Comments:**

Dear Authors,

Thank you for submitting your manuscript.

After editorial evaluation and consideration of the reviewer’s comments, major revision is required before the manuscript can be reconsidered for publication. Please prepare a point-by-point response, addressing each reviewer comment individually and explicitly. For every comment, clearly indicate how the manuscript has been revised, including the location of changes, or provide a concise justification if no revision was made.Responses should be clearly numbered to correspond to the reviewer’s comments.

The reviewer has suggested citation of a specific reference.Inclusion of this reference is not mandatory, and the decision is left to the authors’ discretion. However, if you consider the reference relevant to the rationale, methodology, or interpretation of your study, please consider citing it and explaining its relevance.

We look forward to receiving your revised manuscript together with a detailed response to the reviewer.

Sincerely,

Kazunori Nagasaka

Reviewers' comments:

Reviewer's Responses to Questions

**Comments to the Author**

1. If the authors have adequately addressed your comments raised in a previous round of review and you feel that this manuscript is now acceptable for publication, you may indicate that here to bypass the “Comments to the Author” section, enter your conflict of interest statement in the “Confidential to Editor” section, and submit your "Accept" recommendation.

Reviewer #3: (No Response)

2. Is the manuscript technically sound, and do the data support the conclusions?

Reviewer #3: Partly

3. Has the statistical analysis been performed appropriately and rigorously? 

Reviewer #3: No

4. Have the authors made all data underlying the findings in their manuscript fully available?

Reviewer #3: No

5. Is the manuscript presented in an intelligible fashion and written in standard English?

Reviewer #3: Yes

6. Review Comments to the Author

Reviewer #3: Tittle page

 Authors are advised to review the journal’s author guidelines and follow a classic structure when submitting manuscripts.

Abstract

 A structured abstract is recommended.

 Establish cut-off points for anemia and iron deficiency for Hb, Ferritin, TSAT.

 The abstract refers to a strong correlation; however, the corresponding quantitative values and p-values are not reported.

Introduction

 Lines 76 to 78 contain references that are not up to date. I recommend incorporating the following reference, which is more relevant to the area of research:

Gonzales GF, Aguilar J, Yucra S, Vásquez-Velásquez C. A novel diagnostic approach to differentiate iron overload from inflammation in children using transferrin saturation (TSAT) and ferritin-based indices: A cross-sectional study. Sci Prog. 2025 Oct-Dec;108(4):368504251385071. doi: 10.1177/00368504251385071.

 A reference should be provided to support the statements in lines 79–82.

Methodology

 The study design is not clearly described. Although the authors indicate that the analysis is based on secondary data, this does not necessarily classify the study as retrospective.

 Definitions should be accompanied by appropriate references.

 In the statistical analysis section, please specify the measure of dispersion reported alongside the median.

 Evaluation of diagnostic validity would benefit from the use of logistic regression modeling and receiver operating characteristic (ROC) curve analysis. Alternatively, the bias-corrected Kappa index may be considered.

 The section on ethical considerations should be placed after the analysis plan, and should first emphasize the study design.

7. PLOS authors have the option to publish the peer review history of their article (what does this mean?). If published, this will include your full peer review and any attached files.

Reviewer #3: No

---

## [Author Response · Author response to Decision Letter 2]

2 Feb 2026

Response to Reviewers

Title page

 Authors are advised to review the journal’s author guidelines and follow a classic structure when submitting manuscripts.

Response: Thank you for this suggestion. We reviewed the author guidelines of PLOS ONE and revised the manuscript layout to follow a standard structure.

Abstract

 A structured abstract is recommended.

Response: Following your recommendation, we agree have structured the abstract using the following subheadings: Background, Methods, Results, and Conclusions.

 Establish cut-off points for anemia and iron deficiency for Hb, Ferritin, TSAT.

Response: Thank you for this comment. We have now explicitly stated all prespecified cut-off values in the structured Abstract and the Definitions subsection of Methods. We defined anemia as Hb <11 g/dL to focus on clinically significant (moderate-to-severe) anemia, consistent with the WHO severity classification for non-pregnant women and the commonly used criteria in Japanese gynecologic practice.

 The abstract refers to a strong correlation; however, the corresponding quantitative values and p-values are not reported.

Response: We agree with your observation. We have added the relevant correlation coefficients and p-values in the Results subsection of the structured abstract to support the stated interpretation.

Introduction

 Lines 76 to 78 contain references that are not up to date. I recommend incorporating the following reference, which is more relevant to the area of research:

Gonzales GF, Aguilar J, Yucra S, Vásquez-Velásquez C. A novel diagnostic approach to differentiate iron overload from inflammation in children using transferrin saturation (TSAT) and ferritin-based indices: A cross-sectional study. Sci Prog. 2025 Oct-Dec;108(4):368504251385071. doi: 10.1177/00368504251385071.

Response: We appreciate this suggestion. We revised the statement on ferritin cut-offs in the Introduction and updated the supporting references by prioritizing recent guideline sources (e.g., WHO 2020; BSG 2021). Additionally, we incorporated the recommended study by Gonzales et al. (2025), discussing TSAT- and ferritin-based interpretation of iron status in the context of inflammation.

 A reference should be provided to support the statements in lines 79–82.

Response: Thank you for this helpful comment. We agree and have added appropriate references to support the description of TSAT as an indicator of iron availability (including the <20% threshold) and its calculation based on serum iron and total iron-binding capacity (TIBC).

Methodology

 The study design is not clearly described. Although the authors indicate that the analysis is based on secondary data, this does not necessarily classify the study as retrospective.

Response: Thank you for your suggestion. We clarified the study design by explicitly describing this work as a retrospective, cross-sectional, observational study based on electronic medical records from the first outpatient visit.

 Definitions should be accompanied by appropriate references.

Response: We appreciate this helpful comment. We added appropriate references to the Definitions subsection to support each operational definition and cut-off (Hb, ferritin, TSAT, and functional/overall iron deficiency).

 In the statistical analysis section, please specify the measure of dispersion reported alongside the median.

Response: Thank you for your suggestion. We clarified that continuous variables were summarized as medians (range) and categorical variables as n (%) in the Statistical analysis subsection.

 Evaluation of diagnostic validity would benefit from the use of logistic regression modeling and receiver operating characteristic (ROC) curve analysis. Alternatively, the bias-corrected Kappa index may be considered.

Response: Thank you for your thoughtful suggestion. Among patients with ferritin measured at the first visit (n=124), agreement between ferritin <20 ng/mL and TSAT <20% was fair (Cohen’s κ=0.375), with similar PABAK values (0.371). We performed multivariable logistic regression for TSAT-only classification (TSAT <20% with ferritin ≥20 ng/mL), adjusting for age and hypermenorrhea. Adenomyosis showed a trend toward higher odds compared with no uterine structural abnormality (OR 3.123, 95% CI 0.942–11.30; p=0.0688), whereas myoma was not associated (OR 1.234, 95% CI 0.399–4.079; p=0.7184). Model discrimination was modest (AUC=0.678, 95% CI 0.564–0.792; ROC in S1 Fig). Relevant text has been added to Methods and Results.

 The section on ethical considerations should be placed after the analysis plan, and should first emphasize the study design.

Response: We agree with your thoughtful comment. We reorganized the Methods section, placing the study design and statistical analysis before ethical considerations. Specifically, we placed the “Ethic

---

## [Decision Letter · Decision Letter 2]

10 Feb 2026

PONE-D-25-29781R2
Characteristics of transferrin saturation and anemia-related biomarkers in patients with uterine adenomyosis
PLOS One

Dear Dr. Ito,

Thank you for submitting your manuscript to PLOS ONE. After careful consideration, we feel that it has merit but does not fully meet PLOS ONE’s publication criteria as it currently stands. Therefore, we invite you to submit a revised version of the manuscript that addresses the points raised during the review process.

We look forward to receiving your revised manuscript.

Kind regards,

Kazunori Nagasaka

Academic Editor

PLOS One

**Journal Requirements:**

**Additional Editor Comments:**

Dear Authors,

We apologize for the delay in the review process and for any inconvenience this may have caused.

The reviewer has provided several comments; however, only one point requires your attention at this stage. According to the journal’s guidelines, the incorporation of coded data into an appropriate public repository is indicated. Please revise the Data Availability Statement accordingly and ensure compliance with the PLOS ONE data policy.

No other substantive revisions are required. We look forward to receiving your revised manuscript.

Sincerely,

Kazunori Nagasaka

Reviewers' comments:

Reviewer's Responses to Questions

**Comments to the Author**

1. If the authors have adequately addressed your comments raised in a previous round of review and you feel that this manuscript is now acceptable for publication, you may indicate that here to bypass the “Comments to the Author” section, enter your conflict of interest statement in the “Confidential to Editor” section, and submit your "Accept" recommendation.

Reviewer #3: All comments have been addressed

2. Is the manuscript technically sound, and do the data support the conclusions?

Reviewer #3: Partly

3. Has the statistical analysis been performed appropriately and rigorously? 

Reviewer #3: Yes

4. Have the authors made all data underlying the findings in their manuscript fully available?

Reviewer #3: No

5. Is the manuscript presented in an intelligible fashion and written in standard English?

Reviewer #3: Yes

6. Review Comments to the Author

**Reviewer #3:** According to the journal's guidelines, the incorporation of coded data into a repository is indicated.

7. PLOS authors have the option to publish the peer review history of their article (what does this mean?). If published, this will include your full peer review and any attached files.

Reviewer #3: No

---

## [Author Response · Author response to Decision Letter 3]

15 Feb 2026

The reviewer has provided several comments; however, only one point requires your attention at this stage. According to the journal’s guidelines, the incorporation of coded data into an appropriate public repository is indicated. Please revise the Data Availability Statement accordingly and ensure compliance with the PLOS ONE data policy.

No other substantive revisions are required. We look forward to receiving your revised manuscript.

Response:

Thank you for giving us the opportunity to revise our manuscript.

We have uploaded our raw dataset to a public repository and revised the Data Availability Statement in the manuscript.

---

## [Decision Letter · Decision Letter 3]

26 Feb 2026

Characteristics of transferrin saturation and anemia-related biomarkers in patients with uterine adenomyosis

PONE-D-25-29781R3

Dear Dr. Ito,

We’re pleased to inform you that your manuscript has been judged scientifically suitable for publication and will be formally accepted for publication once it meets all outstanding technical requirements.

Kind regards,

Kazunori Nagasaka

Academic Editor

PLOS One

Additional Editor Comments (optional):

Dear Dr. Fumitake Ito,

I am pleased to inform you that your manuscript is accepted for publication in PLOS ONE.

We appreciate your patience throughout what has been a longer-than-usual review process.

Thank you for your thoughtful revisions and continued engagement during this time.

Your manuscript will now proceed to the production stage.

The production team will contact you regarding proofs and any additional publication requirements.

Thank you for choosing PLOS ONE for the dissemination of your work.

Sincerely,

Kazunori Nagasaka

Editor

PLOS One

Reviewers' comments:

Reviewer's Responses to Questions

**Comments to the Author**

1. If the authors have adequately addressed your comments raised in a previous round of review and you feel that this manuscript is now acceptable for publication, you may indicate that here to bypass the “Comments to the Author” section, enter your conflict of interest statement in the “Confidential to Editor” section, and submit your "Accept" recommendation.

Reviewer #3: All comments have been addressed

2. Is the manuscript technically sound, and do the data support the conclusions?

Reviewer #3: Partly

3. Has the statistical analysis been performed appropriately and rigorously? 

Reviewer #3: Yes

4. Have the authors made all data underlying the findings in their manuscript fully available?

Reviewer #3: Yes

5. Is the manuscript presented in an intelligible fashion and written in standard English?

Reviewer #3: Yes

6. Review Comments to the Author

Reviewer #3: (No Response)

7. PLOS authors have the option to publish the peer review history of their article (what does this mean?). If published, this will include your full peer review and any attached files.

Reviewer #3: No

---

## [Editor Report · Acceptance letter]

PONE-D-25-29781R3

PLOS One

Dear Dr. Ito,

I'm pleased to inform you that your manuscript has been deemed suitable for publication in PLOS One. Congratulations! Your manuscript is now being handed over to our production team.

Kind regards,

on behalf of

Professor Kazunori Nagasaka

Academic Editor

PLOS One